# Long-term protection of HPV test in women at risk of cervical cancer

Raquel Ibáñez[1]*, Esther Roura[1,2], Laura Monfil[1], Luís Alejandro Rodríguez[3], Montserrat Sardà[4], Nàyade Crespo[5], Amparo Pascual[5], Clara Martí[6], Montserrat Fibla[7], Cristina Gutiérrez[8], Belén Lloveras[9], Gloria Oliveras[10], Anna Torrent[11], Isabel Català[12], Francesc Xavier Bosch[1], Laia Bruni[1], Silvia de Sanjosé[2,13]

**1** Unit of Infections and Cancer—Information and Interventions; Cancer Epidemiology Research Programme, IDIBELL, Catalan Institute of Oncology (ICO), L'Hospitalet de Llobregat, Barcelona, Spain, **2** Centro de Investigación Biomédica en Red: Epidemiología y Salud Pública (CIBERESP CB06/02/0073), Madrid, Spain, **3** Public Health Master, Pompeu Fabra university, Barcelona, Spain, **4** Pathology Department, Consorci Hospitalari de Vic, Vic, Barcelona, Spain, **5** Sexual and Reproductive Health centre of Bages-Solsonès, Institut Català de la Salut, Manresa, Barcelona, Spain, **6** Pathology Department, Hospital General de Granollers, Granollers, Barcelona, Spain, **7** Pathology Department, Hospital Universitari Joan XXIII de Tarragona, Tarragona, Spain, **8** Clinical Laboratory ICS Tarragona, Molecular Biology Section, Hospital universitari Joan XXIII de Tarragona, IISPV Rovira i Virgili University, Tarragona, Spain, **9** Pathology Department, Hospital del Mar, Barcelona, Spain, **10** Pathology Department, Hospital universitari Dr, Josep Trueta de Girona, Catalan Institute of Oncology, Girona, Spain, **11** Sexual and Reproductive Health centre of Mollet del Vallés, Institut Català de la Salut Mollet del Vallès, Barcelona, Spain, **12** Pathology Department, Hospital universitari de Bellvitge, IDIBELL, Catalan Institute of Oncology, L'Hospitalet de Llobregat, Barcelona, Spain, **13** PATH, Seattle, Washington, United States of America

* raquelip@iconcologia.net

**Data Availability Statement:** These are sensitive data that are part of the evaluation activities for cervical cancer screening in Catalonia commissioned by the Oncology Master Plan and the Health Department of the Generalitat of

## Abstract

### Objective

To evaluate the 9-year incidence of cervical intraepithelial neoplasia grade 2 or worse (CIN2 +) and cumulative adherence to perform a next test in a cohort of women aged 40+ years with no cervical screening cytology within a window of 5 years (underscreened women), after baseline cervical cytology and HPV tests.

### Methods

In Catalonia, Spain, co-testing with cytology and HPV test has been recommended in the Public Health system since 2006 for underscreened women. In 2007, 1,594 women with underscreened criteria were identified and followed through medical records form Pathological Department. 9-year cumulative incidence of histologically confirmed CIN2+ and cumulative adherence to perform a next test were estimated using Kaplan-Meier statistics.

### Results

Follow-up was available for 1,009 women (63.3%) resulting in 23 women with. CIN2+ (2.3%). Of them, 4 women (17%) had both tests negative at baseline (3CIN2 and 1CIN3) with cumulative incidence of CIN2+ of 0.4% (95% CI: 0.1–1.4) at 5-years and 1.3% (95% CI: 0.4–3.7) at 9-years. During the first year, the prevalence among women with both tests positive was 27.0% (95% CI: 13.0–50.6) for CIN2+. Lost to follow-up was higher among women

Catalonia. Requests to access this data can be addressed to Josep Alfons Espinàs, from the Oncology Master Plan, using the following email address: ja.espinas@iconcologia.net, arguing the reason why they are requested.

**Funding:** The funders had no role in study design, data collection and analysis, decision to publish, or preparation of the manuscript. This work was partially supported by grants from: LB: Agència de Gestió d'Ajuts Universitaris i de Recerca of the Government of Catalonia (Grant number 2017 SGR 1718). RI: AEPCC investiga 2016 – Xavier Castellsagué Grant. Finally this study has been funded by Instituto de Salud Carlos III through the project PI16/01254 and CIBERESP CB06/02/0073 (co-funded by FEDER funds/European Regional Development Fund (ERDF), a way to build Europe) and RecerCaixa 2015 (MD088652). We also thank CERCA Programme/Generalitat de Catalunya for institutional support. The funders had no role in study design, data collection and analysis, decision to publish, or preparation of the manuscript.

**Competing interests:** Cancer Epidemiology Research Programme runs some research projects funded by MSD and Seegene. Raquel Ibáñez has collaborated in a consultancy for Hologic and received HPV tests free of charge by Roche. Francesc Xavier Bosch has received travel/speaking grants from MSD, Seegene and Roche. The rest of the authors reported having no personal conflict of interest. This does not alter our adherence to PLOS ONE policies on sharing data and materials.

with both tests negative compared to those with both positive tests (38.7% vs 4.2%, p-value <0.001). 40.5% of the women HPV-/cyto- had a re-screening test during the 4 years following the baseline, increasing until 53.5% during the 6 years of follow-up.

## Conclusions

HPV detection shows a high longitudinal predictive value at 9-year to identify women at risk to develop CIN2+. The data validate a safe extension of the 3-year screening intervals (current screening interval) to 5-year intervals in underscreened women that had negative HPV result at baseline. It is necessary to establish mechanisms to ensure screening participation and adequate follow-up for these women.

## Introduction

The implementation of screening programs for cervical cancer prevention, using cervical cytology with a high coverage, has led to a reduction in the incidence and mortality of cervical cancer up to 80%, due to the detection and treatment of precancerous lesions and incipient cancers [1–3]. Unfortunately, this success has been mainly limited to high resource settings.

The recognition of the viral aetiology of cervical cancer has allowed the design of new approaches for the screening of the disease with the aim of improving cytological screening programs [4–6]. Data from randomized clinical trials (RCTs) have demonstrated the superiority of HPV test for the detection of cervical intraepithelial neoplasia grade 2 or worse (CIN2+) or cervical intraepithelial neoplasia grade 3 or worse (CIN3+) [7–11]. The combination of both tests, cervical cytology and HPV tests, results in minor and statistically insignificant improvements in sensitivity for CIN2+ and CIN3+ compared to HPV testing alone [12]. Four of these clinical trials conducted in European countries and for which there are follow-up data for at least two screening rounds (POBASCAM, SWEDESCREEN, ARTISTIC, and NTCC) also revealed a substantial reduction of invasive cervical cancer in the HPV arm. This protective effect was observed starting 2.5 years after baseline in the total group of women screened in both arms and increased over time [13].

However, the absence of screening, ranging from in 23 to 71%, is associated with an increased mortality [14–18]. Our previous data indicated that 3 out of 4 cervical cancer cases in Catalonia (Spain) arise from unscreened women, in agreement with other reports from the country [19,20].

Aiming to reduce these preventable cases, the Autonomic Catalan government approved in June 2006 a cervical cancer screening guidelines, aiming to identify women with suboptimal screening history. This screening program was opportunistic. The guidelines recommended co-testing with HPV DNA and cervical cytology in women aged 40 years or older having no previous screening registered test in a window of 5 years, so called underscreened women [21]. In 2007, as part of the Catalan Institute of Oncology surveillance activities, a follow-up study was initiated in a cohort underscreened women.

The objective of this study was to estimate the predictive value of the co-test of histologically confirmed CIN2+ lesions. A previous analysis evaluating 3-year follow-up showed that this strategy identified women at risk of CIN2+ that would have been missed by using cytology alone [22]. We have extended now the follow-up to up nine years. Also, we evaluated cumulative adherence to perform a next test in this group of women.

## Methods

### Study population

This study is part of periodical evaluation reports on cervical cancer screening activities in the public health system in Catalonia (Spain) carried out by the Catalan Institute of Oncology [22,23].

This retrospective cohort includes 1,831 women with underscreened criteria identified in 2007. These women were offered a co-testing with cytology and HPV test at baseline and had been follow-up for subsequent screening visits up to 2016. If the baseline result was negative for both tests, a subsequent cytology test, as guidelines at the time recommended, was indicated every 3 years until the age of 65 [21]. When cytology was positive, women were directly referred to colposcopy. In HPV+/cyto- women, cytology was recommended at 6 moths. If this cytology was negative, performing HPV test was recommended at 12 months, but if cytology was positive women were referred to colposcopy evaluation.

The overall project was approved by the ethical committee of the Bellvitge University Hospital (PR271/11). Any information regarding the identification of patients was anonymized before analysis.

### Follow-up

Follow-up was available through the computerized clinical history of the pathology laboratories of the reference hospitals. During the follow-up period (2007–2016), date and results of subsequent cytologies, biopsies and HPV test registered in clinical records were collected for all participants. The participant laboratories included were: Hospital Universitari Dr. Josep Trueta, Consorci hospitalari de Vic, Hospital Universitari Joan XXIII, Hospital del Mar, Hospital Universitari de Bellvitge, Hospital General de Granollers, Hospital d'Althaia and Hospital Germans Trias i Pujol.

If further concomitant tests were available, the highest cytological or histological grade of abnormality was taking into account. Women with no more tests apart of those obtained at the baseline were considered "lost to follow-up".

Clinical records on pathology diagnosis and gynecological visits registered from 1st January 2007 to 31st of December 2016 were retrieved regularly every 18 months approximately.

### Screening tests and gold standard test

Cervical cytology was largely based on conventional Pap smears and HC2 transport media was used for HPV evaluation. If liquid based cytology was available, HPV and cytology tests were performed using the same sample. All cytological results were classified according to the 2001 Bethesda system by the institution's pathology teams [24]. Abnormal or positive cytology was defined as atypical squamous cell of undetermined significance (ASC-US) or more severe cytological diagnosis.

The HPV test was performed with the Hybrid Capture 2 test technology (HC2; Qiagen, Gaithersburg, MD, USA) which detects 13 high-risk HPV types: 16, 18, 31, 33, 35, 39, 45, 51, 52, 56, 58, 59 and 68. A positive result was considered if attained or exceeded the FDA-approved threshold of 1.0 relative light unit (RLU/CO). All HPV reference laboratories were part of a quality control with kappa values over 90% [25].

All abnormal cytology results at the level of HSIL+ had to be histologically confirmed and were classified according to the CIN classification [26].

## Statistical analysis

All women identified during 2007 with underscreened criteria, who had been screened by co-test were included. Included laboratories were health area referent for cytology, HPV and histology data and were able to provide follow-up information on the study population.
    We estimated:

1. **Cumulative and annual incidence of CIN2+ and CIN3+.** The main event of interest was the development of CIN2+ or CIN3+ along the follow-up period. CIN2+ result included all histologically confirmed diagnoses of CIN2, CIN3, adenocarcinoma and invasive cervical carcinoma. In CIN3+ result, CIN2 is not included. A variable called "final diagnosis" was created including three categories: 1) women who had developed a CIN2+ or CIN3+ throughout the follow-up period, 2) women with a diagnosis other than CIN2+ or CIN3+, and 3) women with only the baseline tests that were considered lost to follow-up because of missing data. A woman was censored for follow-up if a CIN2+ lesion was identified, or when there was a surgical treatment for a CIN1 lesion, or in case of hysterectomy for non-cervical causes. In these cases, the woman contributed time until the diagnosis of CIN2+ or until time of surgical treatment or hysterectomy. Remaining women or in case of death for causes other than cervical cancer, the contributing time was accounted for until the last recorded test. For women with a baseline positive test, at least two negative tests (either HPV or cytology) were necessary to be registered as having a negative final diagnosis. For women with a negative result of the two tests at baseline, they were coded as negative at the end of follow-up if they had subsequent negative test results.

2. **Cumulative adherence to perform a next test**. Among women with HPV-/cyto- at baseline, any test performed after 180 days following baseline tests including HPV test, cytology or cervical biopsy were all taken into account. This was done to avoid including additional tests not related to screening that could have been performed before (e.g.: follow-up of infections, such as *Candia Albicans*, *Trichomona Vaginalis*, etc.). Among women HPV+/cyto-, tests performed at least 45 days after baseline were considered to not interfere with other tests performed concomitantly with the baseline ones. Among women HPV-/cyto+ and HPV+/cyto+, all subsequent tests performed, after baseline ones were considered without restriction.

    A woman was censored at the time that she had the next cytology, HPV test or cervical biopsy (event) registered after baseline according to the established criteria. The contribution time of that woman was time between baseline tests and the performance of next test. Women did no additional tests were censored on the date of baseline. In case of hysterectomy not due to cervical cancer, or death not related to cervical cancer, women was censored at the date of that event and considered of non-related event. If no date of death was available, the last date of woman's test was used. If none was available, it was considered the date of baseline.

    Results for continuous variables were given as mean values, and for categorical variables as percentages. We used chi-square test to compare proportions between groups. The percentage of cumulative incidence of histologically confirmed CIN2+ and CIN3+ lesions and its 95% confidence intervals (CI) as well as the cumulative adherence to perform a next test were estimated and graphically represented using the Kaplan-Meier curves at 1, 3, 5 and 9 years. Data are presented according to the combination of baseline results of cytology and HPV tests up to 9 years follow-up. Incidence rates were calculated in person-years per 100 women. Pairwise comparisons between cumulative incidence curves were evaluated using log-rank test. Also, annual incidence rates of CIN2+ and CIN3+ lesions were calculated at 3, 5 and 9 years, as the number of cases per 100 women/year, according to all combinations of baseline test results,

with their 95% CI (S1 Table). In all cases, time was expressed in years. All statistical tests were two tailed, and p-values below 0.05 were considered statistically significant.

To estimate number of women at risk during the study period and accounting for variability, person-years were estimated as number of years from baseline date of initial screening tests (HPV/cytology) to date of censorship or until end of follow-up period.

Data analyses were carried out with R software version 3.3.2 (R Core Team (2015). R: A language and environment for statistical computing. R Foundation for Statistical Computing, Vienna, Austria. URL https://www.R-project.org/).

## Results

At baseline1,831 women were included in the study, with a mean age of 54.4 years old at baseline (range 40–88). Double negative tests results were detected in 92.4% of women (n = 1,692) while 1.3% (n = 24) had both tests positive. 6.7% (n = 123) of women were HPV positive irrespective of cytology result, and 2.2% (n = 40) had a positive cytology regardless of HPV result (Fig 1).

At baseline, 231 women out of 1,831 were over 65 years of age with negative result in both tests (HPV/cytology) and were informed, consequently, to stop further cervical cancer screening. Six women died for reasons other than cervical cancer before the performance of a new test, so they were excluded, leaving a total of 1,594 women for being followed-up. Of these,

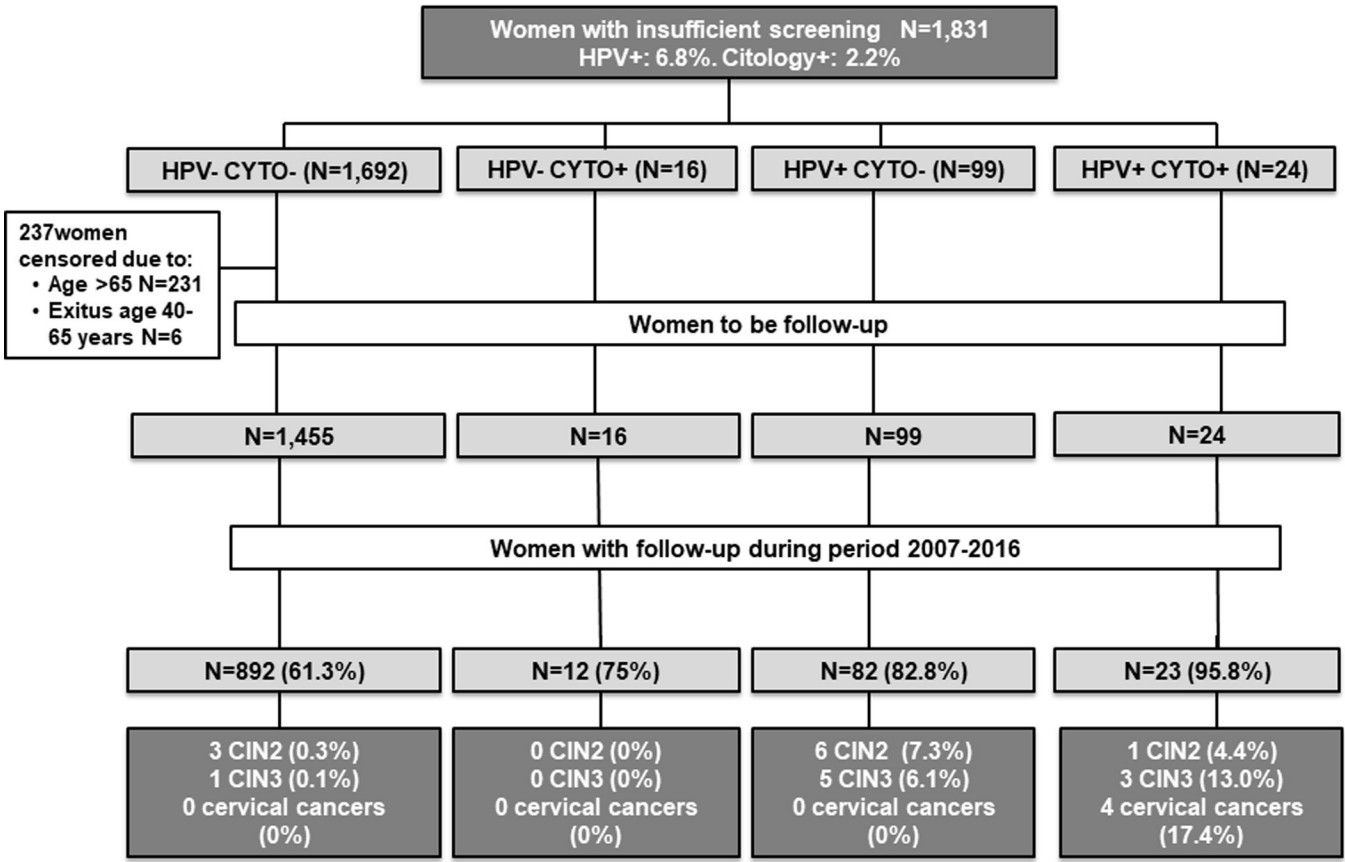

**Fig 1. Flowchart of the study population.** Underscreened women are defined as women older than 39 years with no records on cervical cytology during the previous 5 years.

1,009 (63.3%) women had a complete follow-up during the study period (Fig 1). In 36.7% of women, there was no evidence of a new test during the 9-year follow-up period. Most of these women classified as "lost to follow-up" were women with both baseline negative tests instead of women with both positive results (38.7% vs. 4.2%, p-value <0.001).

During the 9 years of follow-up, a total of 23 women with histologically confirmed CIN2 + lesions were diagnosed, including 9 CIN3, 3 squamous carcinomas and one adenocarcinoma. The mean age of women at CIN2+ diagnosis was 49.3 years (range 40–67 years old). Eleven women (47.8%) out of the 23 CIN2+ cases were diagnosed among women with negative cytology but HPV positive test (Fig 1). Four cases of CIN2+ were diagnosed among women with both negative tests: 3 CIN2 and 1 CIN3. The four cervical cancers (an adenocarcinoma stage IA and 3 squamous carcinoma all of them stage IIB) were diagnosed among women with both positive cytology and HPV results. The average time between first abnormal test and final cancer diagnosis was 2.7 years (range: 0.0–8.4 years).

Among CIN2+ cases, 70% (n = 16) were diagnosed within the first 3 years of follow-up, and all of them were positive for HPV, but only 7 had an abnormal cytology. Of the remaining 30% of cases (n = 7) diagnosed after 3 years of follow-up, one case had an abnormal cytology (14.3%) and the HPV test identified 3 cases (43%). Among women with both tests being negative, 4 had a CIN2+ with an average time until diagnosis of 5.7 years (range of 4.4–7.5 years).

Table 1 shows the cumulative incidences estimates of histologically confirmed CIN2+ and CIN3+ by baseline screening tests results and time period of follow-up. Remarkably, the cumulative incidence of CIN2+ among women with both negative tests was 0.4% (95% CI: 0.1–1.4) at 5 years and 1.3% (95% CI: 0.4–3.7) at 9 years, while, the incidence of CIN3+ was zero in the first 5 years of follow-up and 0.3% (95% CI: 0.0–2.2) at 9 years. Among women with both positive tests at baseline, the cumulative incidence were 27.0% (95% CI: 13.0–50.6) during the first year for CIN2+ and 21.7% (95% CI: 9.7–44.6) for CIN3+, increasing up to 32.6% (95% CI: 16.9–56.9) and 27.8% (95% CI: 13.3–52.2) for CIN2+ and CIN3+ respectively at 3 and 5 years.

Annual incidence rates of CIN2+ and CIN3+ lesions at 3, 5 and 9 years were detailed in the supporting information (S1 Table).

Fig 2 summarizes graphically the cumulative risk of developing CIN2+ and CIN3+ lesions over time for the different combinations of test results at baseline. Among all women with a positive cytology, only those with an HPV positive test developed CIN2+ or CIN3+. The cumulative risk of developing CIN3+ lesions among those women with a negative HPV test remained very low throughout the 9 years of follow-up. The risk of developing CIN3+ among negative HPV women regardless of cytology result at 5 years was zero (Table 1).

Differences between the combination of screening test results and having a CIN2+ or CIN3 + lesion were statistically significant according to the log-rank test (p-value <0.001) an explained by the impact of HPV test and not by cytology results. When HPV negative women with negative and positive cytology were compared, these differences were not statistically significant (p-value = 0.8 for CIN2+ and p-value = 0.9 for CIN3+). Thus, HPV test was crucial in the CIN2+ risk stratification.

Table 2 and Fig 3 show the cumulative adherence to perform a next cytology, HPV test or cervical biopsy, along time by results of baseline screening tests. We observed that among women with both negative tests at baseline, in which the recommended interval for re-screening was after 3 years, only 40.5% had a next test registered during the 4 years from the baseline and 53.5% returned to perform a test along the following 6-years after baseline. Only 60.2% of these women had a next test recorded during the 9-years follow-up period, evidencing a poor adherence of these women to screening protocol recommendations. According to screening protocol recommendations, in HPV+/cyto- women, a cytology and colposcopy should be

**Table 1. Cumulative incidence of histologically confirmed CIN2+ and CIN3+ lesions among underscreened women by baseline co-testing results.**

| Screening tests results | Number of women at risk | Number of CIN2+ | Number of women censored | Cumulative CIN2+ incidence (%) | 95% Conf. Interval (%) | | Number of CIN3+ | Number of women censored | Cumulative CIN3+ incidence (%) | 95% Conf. Interval (%) | |
|---|---|---|---|---|---|---|---|---|---|---|---|
| **HPV positive—cyto positive (N = 24)** | | | | | | | | | | | |
| **Period of time (years)** | | | | | | | | | | | |
| 1 | 14 | 6 | 4 | 27.0 | 13.0 | 50.6 | 5 | 5 | 21.7 | 9.7 | 44.6 |
| 3 | 7 | 1 | 6 | 32.6 | 16.9 | 56.9 | 1 | 6 | 27.8 | 13.3 | 52.2 |
| 5 | 5 | 0 | 2 | 32.6 | 16.9 | 56.9 | 0 | 2 | 27.8 | 13.3 | 52.2 |
| 9 | . | . | . | . | . | . | . | . | . | . | . |
| **HPV positive—cyto negative (N = 99)** | | | | | | | | | | | |
| **Period of time (years)** | | | | | | | | | | | |
| 1 | 65 | 5 | 29 | 6.5 | 2.8 | 14.9 | 3 | 31 | 4.1 | 1.3 | 12.2 |
| 3 | 38 | 4 | 23 | 14.0 | 7.4 | 25.6 | 1 | 26 | 5.8 | 2.2 | 14.9 |
| 5 | 20 | 1 | 17 | 17.1 | 9.2 | 30.4 | 0 | 18 | 5.8 | 2.2 | 14.9 |
| 9 | 1 | 1 | 18 | 24.6 | 12.0 | 46.5 | 1 | 18 | 14.4 | 4.3 | 42.2 |
| **HPV negative—cyto positive (N = 16)** | | | | | | | | | | | |
| **Period of time (years)** | | | | | | | | | | | |
| 1 | 9 | 0 | 7 | 0.0 | 0.0 | 0.0 | 0 | 7 | 0.0 | 0.0 | 0.0 |
| 3 | 8 | 0 | 1 | 0.0 | 0.0 | 0.0 | 0 | 1 | 0.0 | 0.0 | 0.0 |
| 5 | 6 | 0 | 2 | 0.0 | 0.0 | 0.0 | 0 | 2 | 0.0 | 0.0 | 0.0 |
| 9 | 1 | 0 | 5 | 0.0 | 0.0 | 0.0 | 0 | 5 | 0.0 | 0.0 | 0.0 |
| **HPV negative—cyto negative (N = 1,455)** | | | | | | | | | | | |
| **Period of time (years)** | | | | | | | | | | | |
| 1 | 856 | 0 | 599 | 0.0 | 0.0 | 0.0 | 0 | 599 | 0.0 | 0.0 | 0.0 |
| 3 | 725 | 0 | 131 | 0.0 | 0.0 | 0.0 | 0 | 131 | 0.0 | 0.0 | 0.0 |
| 5 | 498 | 2 | 225 | 0.4 | 0.1 | 1.4 | 0 | 227 | 0.0 | 0.0 | 0.0 |
| 9 | 26 | 2 | 470 | 1.3 | 0.4 | 3.7 | 1 | 471 | 0.3 | 0.0 | 2.2 |

CIN2+ included cervical intraepithelial neoplasia grade 2 and 3 and cervical carcinoma. CIN3+ included cervical intraepithelial neoplasia grade 3 and cervical carcinoma. CIN2+ and CIN3+ incidence rates were calculated in person-years per 100 women.

performed between 6 and 12 months or an HPV test at 12 months as follow-up. We observed that the 67.3% of these women had a follow-up test during the following 2 years from baseline, increasing to 74.5% before 3 years. However, in 16.3% of women there was no additional test recorded during the 9-year follow-up. Most women with a positive cytology at baseline and regardless of the HPV result had a follow-up test in the following year.

## Discussion

HPV testing in cervical cancer screening among underscreened women over age 39 has shown an excellent longitudinal predictive value for the detection of CIN2+ and CIN3+ lesions at 9 years of follow-up, in both HPV positive and HPV negative women. The inclusion of HPV test in addition to cervical cytology in women at risk of cervical cancer identified all CIN3+ cases but one, at year 9 since baseline, while cytology missed 6 cases of CIN3+. These results are consistent with our evaluation at 3 years and with the literature, supporting HPV as a standalone primary screening test [22], and that HPV testing is more accurate than cytology to predict disease also among underscreened women.

Women with HPV negative results, regardless of cytology result had a low risk for CIN2 + and CIN3+, remaining low over time. In our study, no cases with CIN3+ were observed

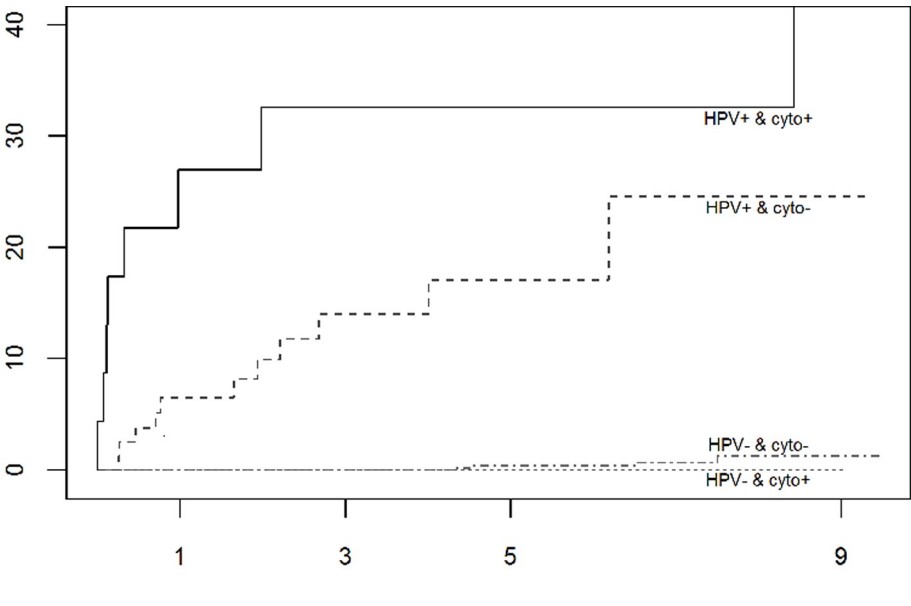

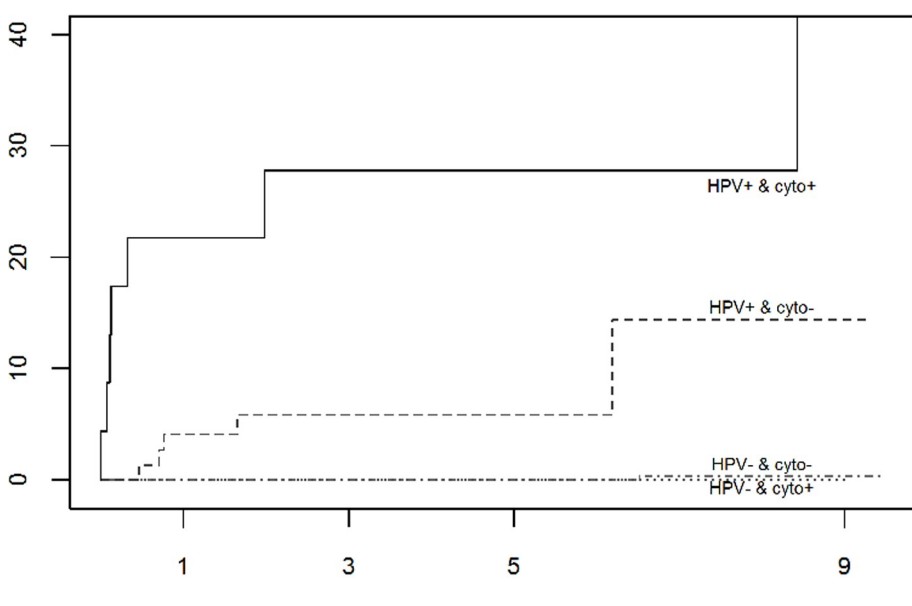

**Fig 2.** Cumulative incidence of developing a histologically confirmed CIN2+ (A) or CIN3+ (B) by baseline co-testing results among underscreened women. Underscreened women are defined as women older than 39 years with no records on cervical cytology during the previous 5 years. CIN2+ included cervical intraepithelial neoplasia grade 2 and 3 and cervical carcinoma. CIN3+ included cervical intraepithelial neoplasia grade 3 and cervical carcinoma.

among HPV negative women at 5 years and the risk increased still a very low level to 0.3% (95% CI, 0.1–2.3%) at 9 years. Additionally, among all women with a positive cytology, only those with a concomitant HPV positive test developed cervical pathology. In the RCT Athena Study, a study conducted in US among women aged 25 years or older attending routine

**Table 2. Cumulative adherence to perform a next cytology, HPV test or cervical biopsy, at different time periods, stratified by baseline co-testing results among underscreened women.**

| Screening tests results at baseline | Number of women at risk | Number of women with a next cyto, HPV test or biopsy | Number of Women censored | Cumulative adherence to perform a next cyto, HPV test or biopsy (%) | 95% Conf. Interval (%) | |
|---|---|---|---|---|---|---|
| **HPV positive - cyto positive (N = 24)** | | | | | | |
| **Period of time (years)** | | | | | | |
| 1 | 2 | 22 | 0 | 91.7 | 76.7 | 98.6 |
| 2 | 1 | 1 | 0 | 95.8 | 82.4 | 99.7 |
| 3 | 1 | 0 | 0 | 95.8 | 82.4 | 99.7 |
| 4 | 1 | 0 | 0 | 95.8 | 82.4 | 99.7 |
| 5 | 1 | 0 | 0 | 95.8 | 82.4 | 99.7 |
| 6 | 1 | 0 | 0 | 95.8 | 82.4 | 99.7 |
| 7 | 1 | 0 | 0 | 95.8 | 82.4 | 99.7 |
| 8 | 1 | 0 | 0 | 95.8 | 82.4 | 99.7 |
| 9 | 1 | 0 | 0 | 95.8 | 82.4 | 99.7 |
| **HPV positive—cyto negative (N = 99)** | | | | | | |
| **Period of time (years)** | | | | | | |
| 1 | 55 | 43 | 1 | 43.9 | 34.7 | 54.3 |
| 2 | 32 | 23 | 0 | 67.3 | 58.0 | 76.4 |
| 3 | 25 | 7 | 0 | 74.5 | 65.6 | 82.6 |
| 4 | 21 | 4 | 0 | 78.6 | 70.0 | 86.1 |
| 5 | 19 | 2 | 0 | 80.6 | 72.3 | 87.7 |
| 6 | 16 | 3 | 0 | 83.7 | 75.7 | 90.2 |
| 7 | 16 | 0 | 0 | 83.7 | 75.7 | 90.2 |
| 8 | 16 | 0 | 0 | 83.7 | 75.7 | 90.2 |
| 9 | 14 | 0 | 2 | 83.7 | 75.7 | 90.2 |
| **HPV negative—cyto positive (N = 16)** | | | | | | |
| **Period of time (years)** | | | | | | |
| 1 | 5 | 10 | 1 | 65.9 | 42.8 | 87.4 |
| 2 | 4 | 1 | 0 | 72.7 | 49.6 | 91.5 |
| 3 | 4 | 0 | 0 | 72.7 | 49.6 | 91.5 |
| 4 | 4 | 0 | 0 | 72.7 | 49.6 | 91.5 |
| 5 | 4 | 0 | 0 | 72.7 | 49.6 | 91.5 |
| 6 | 4 | 0 | 0 | 72.7 | 49.6 | 91.5 |
| 7 | 4 | 0 | 0 | 72.7 | 49.6 | 91.5 |
| 8 | 4 | 0 | 0 | 72.7 | 49.6 | 91.5 |
| 9 | 4 | 0 | 0 | 72.7 | 49.6 | 91.5 |
| **HPV negative—cyto negative (N = 1,455)** | | | | | | |
| **Period of time (years)** | | | | | | |
| 1 | 1,386 | 67 | 2 | 4.6 | 3.6 | 5.8 |
| 2 | 1,230 | 156 | 0 | 15.3 | 13.6 | 17.3 |
| 3 | 1,085 | 145 | 0 | 25.3 | 23.2 | 27.6 |
| 4 | 863 | 220 | 2 | 40.5 | 38.0 | 43.1 |
| 5 | 747 | 116 | 0 | 48.5 | 45.9 | 51.1 |
| 6 | 674 | 73 | 0 | 53.5 | 51.0 | 56.1 |
| 7 | 642 | 32 | 0 | 55.7 | 53.2 | 58.3 |
| 8 | 586 | 49 | 7 | 59.1 | 56.6 | 61.6 |

(*Continued*)

**Table 2.** (Continued)

| Screening tests results at baseline | Number of women at risk | Number of women with a next cyto, HPV test or biopsy | Number of Women censored | Cumulative adherence to perform a next cyto, HPV test or biopsy (%) | 95% Conf. Interval (%) | |
|---|---|---|---|---|---|---|
| 9 | 459 | 14 | 113 | 60.2 | 57.6 | 62.7 |

Underscreened women are defined as women older than 39 years and with no records on cervical cytology during the previous 5 years. Follow-up intervals established according to the protocol are: 1) 3-years among women negative for both tests, 2) between 6 months and 1-year among women positive for HPV but with negative cytology, 3) immediate follow-up using colposcopy for the remaining cases.

cervical screening, the cumulative incidence of CIN3+ after 3 years for women with double negative tests (cytology + HPV) was 0.3% (95% CI, 0.1–0.6%) comparable to that observed among women having been tested with HPV only, and more than twice among women with negative cytology [27]. In another RCT carried out in Sweden, the cumulative incidence of CIN3+ was 0.89% among HPV negative women at 13 years of follow-up, similar to that reported among women with a negative co-test (0.84%), in contrast with a cumulative incidence of 1.54% in women with negative cytology [28]. In the POBASCAM RCT the cumulative CIN3+ incidence was 0.56% among HPV negative women after 3 rounds of screening (14 years follow-up) [29]. Therefore, HPV test allows for a better risk stratification with no clear of co-testing [13,27,28]. Among women with both tests being negative, 4 had a CIN2+ with an average time until diagnosis of 5.7 years (range of 4.4–7.5 years).

On the other hand, in our study, the average time to final diagnosis of a CIN2+ lesion in women with both negative tests at baseline was 5.7 years (range 4.4–7.5) and 6.5 years for a

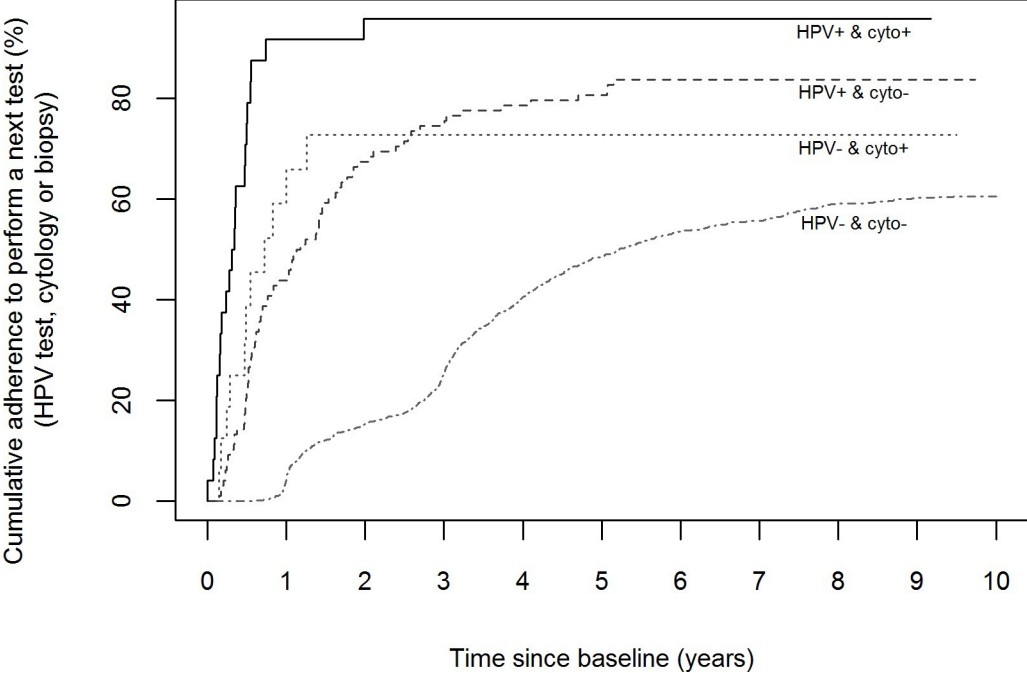

**Fig 3. Cumulative adherence to perform a next test, stratified by baseline co-testing results among underscreened women.** Underscreened women are defined as women older than 39 years and with no records on cervical cytology during the previous 5 years. Follow-up intervals established according to the protocol are: 1) 3-years among women negative for both tests, 2) between 6 months and 1-year among women positive for HPV but with negative cytology, 3) immediate follow-up using colposcopy for the remaining cases.

CIN3+ diagnosis (for the only one case). Our results support that HPV can be an excellent primary test that provides security to go beyond a 3-year screening interval, also in underscreened women. In the European RCTs, risk of CIN3+ among women with a negative screening test at baseline was found to be about the same after six years with HPV screening than after three years with cytology screening [13], suggesting that screening intervals could safely be extended for women who screen with HPV test. Furthermore, the risk of CIN2+ or CIN3+ among HPV negative women was low and maintained the same protection even after 13 years of follow-up [28]. This strong scientific evidence are now being reflected in national guidelines around the world [30–32]. The European guidelines for cervical cancer prevention recommend a screening interval of at least 5 years which may extend up to 10 years for women with a negative HPV primary test result. This range of intervals may vary by age and women's screening history [31].

However, having a screening test with a higher sensitivity but a lower specificity leads to overdetection and needs to be balanced by the benefits of an early detection. To avoid detection of HPV positive cases that are unlikely to develop cervical cancer, several authors have proposed to restrict the number of HPV types included in the HPV assays, given that the risk of developing a CIN3+ is very low for some genotypes such as 39, 51, 56, 59, 68 [33,34]. Recent studies have seen that the risk of developing a 7-year CIN3+ for these types is less than 5% and 80% of these infections are no longer detected in three years [34]. On the other hand, RCTs of HPV based screening have found that the higher detection of CIN2+ at baseline is followed by a reduction of these lesions in subsequent screening rounds [28–30]. These studies showed that the early increases detection of CIN2+ with HPV based screening does not represent a big burden of overdiagnosis but it suggests an earlier detection of cervical disease [13,30]. In our study, we included women above age 39, in whom regressive lesions are likely to be less common as compared to young women[35]. In our study, 86% of CIN2+ diagnosed among women those with both positive tests were detected during the first year and in most cases, screening tests were part of the diagnosis of CIN2+. In addition, having both positive tests generates a much higher risk during the first year than having one of them negative (27.0% vs. 6.5%). But above all, according to the data from our study, the risk increased if the positive test was the HPV test. Subsequently, the risk of diagnosis of these lesions decreases in later years. In fact, in our study, the cumulative risk of having CIN2+ lesions had a steep rise between the first 12 months and the third year, followed by stabilization or slow increase between the third and fifth year of follow-up. RCTs have also shown a reduction in detection of high grade lesions in the second rounds compared with the first round, highlighting the impact of using the HPV test in screening when it is compared with cytological screening [28,29]. This benefit of the HPV test could be critical for underscreened women that have persistently poor adhesion to recommendations. In our study, we found that one third of total women did not return to screening after baseline tests. Among double negative women, only 40.5% had a re-screening before the next 4 years and 53.5% before the 6 years of follow-up, demonstrating that despite having participated in the screening activities, they did not meet the established intervals and most of those who returned, they did so at longer than recommended intervals. Unfortunately, we do not know if these women have searched care elsewhere. Under opportunistic screening approaches, loss to follow-up is common. We have shown that inviting these women can quickly revert the situation [36]. The use of personal reminders increased significantly screening participation even if only an invitation letter with a scheduled visit was used [36,37]. Generally, in organized programs, higher adherence to the screening programs is attained [37–39]. As well, women may perceive themselves to be at low risk of cervical cancer, and thus delay their attendance at screening as occurred in a Norwegian study [40]. A study

carried out in Sweden also pointed out that women have a comprehensive rationale for postponing cervical screening, yet they do not view themselves as non-attenders [41].

On the other hand, a certain proportion of women, 17.2% of HPV positive and negative cytology and in 25% HPV negative women with a positive cytology, had no record of a second screening visit. A poor information or full understanding of a positive test could affect follow-up visit as has been seen elsewhere [42,43]. Good and comprehensive information remains a special need, particularly when the HPV test is positive, but the cytology does not detect any abnormality. The fact of having a negative cytology, associated with a lack of knowledge about the superiority of HPV tests in predicting disease, can give a false sense of security as observed also by others [40,41]. Some studies have shown that attendance at repeated tests is poor, particularly after a normal cytological test, reaching losses of 40% within the established one-year follow-up intervals [44,45]. Adequate communication is necessary to improve adequate management of screen positive women. For women who find it difficult to attend appointments, there may be substantial advantage to being able to self-collect at home.

## Study limitations and strengths

This is an evaluation of the implementation of screening guidelines and data are thus limited to that available at the health information system of public health sector. Any additional information such as socioeconomic profile or individual characteristics that could affect participation in screening activities was not available. The yield of disease among the subgroup of women with HPV-/cyto+, at baseline (only 16 women), should be considered with caution, due to the low number of women under this stratum.

A strong point of our study lies in the monitoring over a period of 9 years the use of pathology laboratory registries. Although there could be some sub-registration if women decided after a given test, to go to a private services, this is however, suspected to be minimal among those having had a baseline visit although no routine registration exist. The HPV test used (HC2) is a robust test with high interlaboratory reproducibility among all reference laboratories in Catalonia [25].

## Conclusions

HPV test in cervical cancer screening, among underscreened women over age 39, shows a high longitudinal predictive value at 9 years to identify women at risk to develop CIN2+, in both HPV+ and HPV- women. The data validate a safe extension of the 3-year screening intervals (current screening interval) to 5-year intervals in underscreened women with a negative HPV result. Underscreened women had persistently poor adhesion to screening recommendations, so it is necessary to establish mechanisms to ensure re-screening participation within the established intervals in women with negative tests and compliance with adequate follow-up in women with positive tests.

## Supporting information

**S1 Table. Annual incidence rate of histologically confirmed CIN2+ and CIN3+ lesions in underscreened women by time periods and results of the baseline screening.** CIN2 + included cervical intraepithelial neoplasia grade 2 and 3 and cervical carcinoma. CIN3 + included cervical intraepithelial neoplasia grade 3 and cervical carcinoma. The CIN2+ and CIN3+ incidence rate was calculated in person-year per 100 women.
(PDF)

## Acknowledgments

We would like to thank all those collaborators involved in collecting follow-up information: R. Redón, L. Murgui, S. Abajo, A. Bragulat and D. Boada.

## Author Contributions

**Conceptualization:** Raquel Ibáñez, Silvia de Sanjosé.

**Data curation:** Raquel Ibáñez.

**Formal analysis:** Raquel Ibáñez, Esther Roura, Laura Monfil, Luís Alejandro Rodríguez.

**Funding acquisition:** Raquel Ibáñez, Francesc Xavier Bosch, Laia Bruni, Silvia de Sanjosé.

**Investigation:** Raquel Ibáñez, Montserrat Sardà, Nàyade Crespo, Amparo Pascual, Clara Martí, Montserrat Fibla, Cristina Gutiérrez, Belén Lloveras, Gloria Oliveras, Anna Torrent, Isabel Català.

**Methodology:** Raquel Ibáñez, Esther Roura, Laura Monfil, Laia Bruni, Silvia de Sanjosé.

**Project administration:** Raquel Ibáñez.

**Resources:** Raquel Ibáñez, Esther Roura, Laura Monfil, Montserrat Sardà, Nàyade Crespo, Amparo Pascual, Clara Martí, Montserrat Fibla, Cristina Gutiérrez, Belén Lloveras, Gloria Oliveras, Anna Torrent, Isabel Català, Laia Bruni, Silvia de Sanjosé.

**Software:** Esther Roura, Laura Monfil.

**Supervision:** Raquel Ibáñez, Silvia de Sanjosé.

**Validation:** Raquel Ibáñez, Esther Roura, Laura Monfil, Luís Alejandro Rodríguez.

**Visualization:** Raquel Ibáñez, Francesc Xavier Bosch, Silvia de Sanjosé.

**Writing – original draft:** Raquel Ibáñez, Esther Roura, Laura Monfil, Luís Alejandro Rodríguez, Laia Bruni, Silvia de Sanjosé.

**Writing – review & editing:** Raquel Ibáñez, Esther Roura, Laura Monfil, Luís Alejandro Rodríguez, Montserrat Sardà, Nàyade Crespo, Amparo Pascual, Clara Martí, Montserrat Fibla, Cristina Gutiérrez, Belén Lloveras, Gloria Oliveras, Anna Torrent, Isabel Català, Francesc Xavier Bosch, Laia Bruni, Silvia de Sanjosé.

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
