## [Decision Letter · Decision Letter 0]

12 May 2020

PONE-D-20-10012

Long-term protection of HPV test in women at risk of cervical cancer

PLOS ONE

Dear Dr. Ibañez,

Thank you for submitting your manuscript to PLOS ONE. After careful consideration, we feel that it has merit but does not fully meet PLOS ONE’s publication criteria as it currently stands. Therefore, we invite you to submit a revised version of the manuscript that addresses the points raised during the review process.

We would appreciate receiving your revised manuscript by Jun 26 2020 11:59PM. To enhance the reproducibility of your results, we recommend that if applicable you deposit your laboratory protocols in protocols.io, where a protocol can be assigned its own identifier (DOI) such that it can be cited independently in the future. For instructions see: http://journals.plos.org/plosone/s/submission-guidelines#loc-laboratory-protocols

We look forward to receiving your revised manuscript.

Kind regards,

Maria Lina Tornesello

Academic Editor

PLOS ONE

Journal Requirements:

2. In the ethics statement in the manuscript and in the online submission form, please provide additional information about the patient records used in your retrospective study, including: a) whether all data were fully anonymized before you accessed them; b) the date range (month and year) during which patients' medical records were accessed; and c) the source of the medical records analyzed in this work (e.g. hospital, institution or medical center name).

3. In the ethics statement in the manuscript and in the online submission form, please provide the ethics approval number that was issued by the ethical committee of the Bellvitge University Hospital."

4. To comply with PLOS ONE submission guidelines, in your Methods section, please provide additional information regarding your statistical analyses, including the specific threshold set to judge statistical significance in your study. For more information on PLOS ONE's expectations for statistical reporting, please see https://journals.plos.org/plosone/s/submission-guidelines.#loc-statistical-reporting.

6. Thank you for stating the following in the Competing Interests section:

"I have read the journal's policy and the authors of this manuscript have the following

competing interests:

RI, ER, LM, LB and FXB: Cancer Epidemiology Research Programme runs some

research projects funded by MSD and Seegene.

RI: has collaborated in a consultancy for Hologic and received HPV tests free of charge

by Roche.

FXB:has received travel/speaking grants from MSD, Seegene and Roche.

The rest of the authors reported having no conflict of interest."

7. Please amend the manuscript submission data (via Edit Submission) to include author BelénLloveras.

8. Please amend your list of authors on the manuscript to ensure that each author is linked to an affiliation. Authors’ affiliations should reflect the institution where the work was done (if authors moved subsequently, you can also list the new affiliation stating “current affiliation:….” as necessary).

Reviewers' comments:

Reviewer's Responses to Questions

**Comments to the Author**

1. Is the manuscript technically sound, and do the data support the conclusions?

Reviewer #1: Partly

Reviewer #2: Yes

Reviewer #3: Yes

2. Has the statistical analysis been performed appropriately and rigorously? 

Reviewer #1: No

Reviewer #2: Yes

Reviewer #3: Yes

3. Have the authors made all data underlying the findings in their manuscript fully available?

Reviewer #1: No

Reviewer #2: Yes

Reviewer #3: Yes

4. Is the manuscript presented in an intelligible fashion and written in standard English?

Reviewer #1: No

Reviewer #2: Yes

Reviewer #3: Yes

5. Review Comments to the Author

Reviewer #1: The paper is interesting as it describes the incidence of cervical diseases (i.e., CIN2+) in a cohort of under-screened women (i.e. no cervical screening in the previous 5 years), who had been invited to screening with both HPV test and Pap Smear test. The follow-up is very long (9 years), therefore, the study results indicate the long protective effect of a negative HPV test, also in under-screened women.

Major comments

- There is a crucial error in Table 1: the cumulative incidence at 9 years is not reported for the HPV+/cyto+ group. It could be a typo, but the value is not reported neither in the text, therefore, I suggest a revision of all the statistical analysis.

- Please uniform the Kaplan-Meier figures with the time span reported in Table 1 (i.e. 1, 3, 5, 9 years in the horizontal axis). Add also censored women (i.e., drop-out without the event of interest) in the curves. It seems strange that the end of the line (last available data) for the 4 groups is not equal to 9 years. Maybe a complete 9-year follow-up has not been reached by some group of women? The time span used for the calculation seems to be months, please specify in the methods.

- Please better specify the difference between censoring for follow-up and censoring as it is usually intended in the Kaplan-Meier estimates. Please describe in details in the methods which are the events of interest: I suppose that HSIL/CIN2+ is an event of interest, but also death from cervical cancer should be one, and hysterectomy for cervical diseases (if possible).

- I wonder if in Spain you are able to distinguish between deaths due to ‘uterus n.o.s.’ and deaths due to cervical cancer. I know this is a challenging issue in several countries. The same questions arise with regard to the possibility of identifying the reason of hysterectomy.

Minor comments

- Please better specify what do you mean for “predictive value”, i.e., both positive and negative predictive values?

- Methods, lines 122-124 death and end of follow-up should be included among the censoring times.

- Please add the total for each group in Table 1.

- The study could be underpowered (from a statistical point of view) to detect incidence of cervical lesions among the subgroup of women with HPV-/Cyto+ at baseline (only 16 women). Therefore, the results (incidence =0) in this group should be considered with caution. Please add this limitation to the discussion.

- Pg. 10 lines 194-196. I do not understand the meaning of this paragraph.

- pg. 11, lines 225-229. I do not understand the meaning this comparison between groups.

Typos/language

Pg. 10: line 203, “3-years stablished” should be “3-year established”; line 212 “anormal”, should be “abnormal”

Pg. 11: Line 215 “CIN2+” should be “HSIL/CIN2+”; line 220 “cero”, should be “zero”; line 225 “statistical significantly” should be “statistically significant”

Pg. 12: Line 240: “baseline” 2 times; Line 241 “an HPV positive tests” should be “a HPV positive test”

Pg. 14: line 273 “accumulative” should be “cumulative”

Pg. 15: line 303 “To avoid detection of positive cases, that are unlikely to develop disease,” should be “To avoid detection of HPV positive cases that are unlikely to develop cervical diseases,”

Pg. 16: line 323 “least”, should be “at least”.

Reviewer #2: The paper is very well written and reports in a sound, original way the results of a cohort including under-screened women tested for HPV and cytology. The small number of women does not allow to infer too much about outcomes in interesting subgroups as cyto+/HPV- women at baseline. There are many other retrospective and prospective studies that analyzed similar cohorts around the world, thus the study can add very little to what is already known on a biological, natural history of the disease and screening effectiveness point of view. Nevertheless, the nature of thee cohort and thee length of follow up makes the study interesting.

I suggest to focus the results and the discussion on the specific issues related to the high risk group and in particular on the compliance to follow up and screening appointments in the following nine years. Now there is a paragraph in the results about timing of rescreening, maybe a figure with survival curves for having a test or the frequency of first and second test during follow up, separately for negative and positive women could be more informative.

Also in the discussion, most of the issues afforded are general and many other studies and secondary literature already has treated these topics. What is specific of this study is the underscreened population over 40. I suggest to discuss points linked to this: I see two issues, but other can be interesting that I do not see: 1) compliance to follow up and acceptability; 2) co-testing is not recommended in Europe, but some researchers argue that for women not screened before 40, where the prevalence of cancer may be high, including cytology in the first screening episode could be useful, given that the cytology vs HPV relative sensitivity for cancer seems to be similar, cytology could add something. For the second point your study cannot give an answer, but you can confirm the high prevalence of cancers in this population, all HPV+/cyto+...

I have some minor comments

Line 193 the lost to follow up seem different from those reported in the abstract.

Line 187 and line 194 the same results are reported with different indicators, thus it gives an impression of inconsistency and it is difficult to follow. I suggest to report the two measures in the same paragraph.

Line 199 the average age at diagnosis is reported, but given the length of follow up it is difficult to manage this information. May be better to say how many in women below and over a certain age and to report it also for the whole cohort.

Lines 225-28: there is something not clear to me in the sentence: “Not statistical significantly differences were detected among women HPV positive and cytology negative at baseline at 9 years and women HPV positive and cytology positive at 5 years (24.6% CI:12.0-46.5; 32.6% CI:16.9-56.9 respectively).” Is the comparison between baseline or 9yy in HPV+/cyto- and HPV+/cyto+ at 5yy?

In general, I suggest to shorten the text of the results: tables and figures are much clearer…

Discussion

See general comments

Reviewer #3: This article attests to the sensitivity of HPV testing for cervical screening and its ability to justify extended screening rounds beyond 3 years. It is well written and the data support the conclusions drawn

Comments

The data support the conclusions drawn although the sample size is relatively small - the authors should avoid overstatement/extrapolation from their data - e.g. while I accept that the risk of developing CIN3 in HPV negative women "regardless of cytology result at 5 years was 0" - this is likely to be influenced by power-limitations

An important element of this work is that the cohort are "underscreened " women - we get to know what is meant by this in the introduction but underscreened is not defined in abstract, and arguably should be

A review of the English is recommended; there are some minor lapses in vocabulary and tenses eg line 83 "arouse" should be "arise" and for line 273 "accumulative" should be "accumulated"

The data will be of importance locally and serve as key audit and service evaluation, with the potential to influence wider policy. However, as the authors describe - there are several larger published studies which describe longitudinal performance of HPV either as a stand alone test or as a co-test with cytology for the detection of cervical disease. The authors should try to describe the novelty/importance of these findings for the international community more clearly

End of review

6. PLOS authors have the option to publish the peer review history of their article (what does this mean?). If published, this will include your full peer review and any attached files.

Reviewer #1: No

Reviewer #2: Yes: paolo giorgi rossi

Reviewer #3: Yes: Catherine Cuschieri

---

## [Author Response · Author response to Decision Letter 0]

14 Jul 2020

Dear editor and reviewers,

Please find enclosed the revised version of the manuscript entitled “Long-term protection of HPV test in women at risk of cervical cancer” for publication, as a research article, in PLoS One.

We have considered all the suggestions provided by the editor and reviewers. We thank them all for this as we believe have contributed to improve our manuscript. Below, you can see our responses point by point. We hope the manuscript is suitable now for publication.

Response to editor: 

Response: It has been revised and adapted to the PLOS ONE's style requirements.

2. In the ethics statement in the manuscript and in the online submission form, please provide additional information about the patient records used in your retrospective study, including: a) whether all data were fully anonymized before you accessed them; b) the date range (month and year) during which patients' medical records were accessed; and c) the source of the medical records analyzed in this work (e.g. hospital, institution or medical center name). 

Response:

a) Any information regarding the personal identification of patients was anonymized before analysis. This is already included in the manuscript in the methods section, at the end of the study population part

b) It was included in the methods section, follow-up part through the next paragraph: “Clinical records on pathology diagnosis and gynaecological visits registered from 1st January 2007 to 31st of December 2016 were retrieved regularly every 18 months approximately”.

c) This information was already included in the methods section, follow-up part. Specifically, it says: “Follow-up was available through the computerized clinical history of the pathology laboratories of the reference hospitals. During the follow-up period (2007-2016), date and results of subsequent cytologies, biopsies and HPV test registered in clinical records were collected for all participants. The participant laboratories included were: Hospital Universitari Dr. Josep Trueta, Consorci hospitalari de Vic, Hospital Universitari Joan XXIII, Hospital del Mar, Hospital Universitari de Bellvitge, Hospital General de Granollers, Hospital d’Althaia and Hospital Germans Trias i Pujol.”

3. In the ethics statement in the manuscript and in the online submission form, please provide the ethics approval number that was issued by the ethical committee of the Bellvitge University Hospital. 

Response: Sorry for not adding the reference before. Please see the approval number PR271/11. It was included in the manuscript in methods section, study population part. 

4. To comply with PLOS ONE submission guidelines, in your Methods section, please provide additional information regarding your statistical analyses, including the specific threshold set to judge statistical significance in your study. 

Response: Please see the additional information included in methods section, statistical analysis part: “Results for continuous variables were given as mean values, and for categorical variables as percentages. We used chi-square test to compare proportions between groups. The percentage of cumulative incidence of histologically confirmed CIN2+ and CIN3+ lesions and its 95% confidence intervals (CI) as well as the cumulative adherence to perform a next test were estimated and graphically represented using the Kaplan-Meier curves at 1, 3, 5 and 9 years. Data are presented according to the combination of baseline results of cytology and HPV tests up to 9 years follow-up. Incidence rates were calculated in person-years per 100 women. Pairwise comparisons between cumulative incidence curves were evaluated using log-rank test. Also, annual incidence rates of CIN2+ and CIN3+ lesions were calculated at 3, 5 and 9 years, as the number of cases per 100 women/year, according to all combinations of baseline test results, with their 95% CI (supplemental material). In all cases, time was expressed in years. All statistical tests were two tailed, and p-values below 0.05 were considered statistically significant”.

5. We note that you have indicated that data from this study are available upon request. PLOS only allows data to be available upon request if there are legal or ethical restrictions on sharing data publicly. In your revised cover letter, please address the following prompts:

b) If there are no restrictions, please upload the minimal anonymized data set necessary to replicate your study findings as either Supporting Information files or to a stable, public repository and provide us with the relevant URLs, DOIs, or accession numbers. 

Response: 

These are sensitive data that are part of the evaluation activities for cervical cancer screening in Catalonia commissioned by the Oncology Master Plan and the Health Department of the Generalitat of Catalonia. Requests to access this data can be addressed to Josep Alfons Espinàs, from the Oncology Master Plan, using the following email address: ja.espinas@iconcologia.net, arguing the reason why they are requested.

6. Thank you for stating the following in the Competing Interests section: "I have read the journal's policy and the authors of this manuscript have the following competing interests: RI, ER, LM, LB and FXB: Cancer Epidemiology Research Programme runs some research projects funded by MSD and Seegene. RI: has collaborated in a consultancy for Hologic and received HPV tests free of charge by Roche. FXB: has received travel/speaking grants from MSD, Seegene and Roche.The rest of the authors reported having no conflict of interest."

Please confirm that this does not alter your adherence to all PLOS ONE policies on sharing data and materials, by including the following statement:

"This does not alter our adherence to PLOS ONE policies on sharing data and materials.” (as detailed online in our guide for authors http://journals.plos.org/plosone/s/competing-interests). If there are restrictions on sharing of data and/or materials, please state these. Please note that we cannot proceed with consideration of your article until this information has been declared.

Please include your updated Competing Interests statement in your cover letter; we will change the online submission form on your behalf. Please know it is PLOS ONE policy for corresponding authors to declare, on behalf of all authors, all potential competing interests for the purposes of transparency. PLOS defines a competing interest as anything that interferes with, or could reasonably be perceived as interfering with, the full and objective presentation, peer review, editorial decision-making, or publication of research or non-research articles submitted to one of the journals. Competing interests can be financial or non-financial, professional, or personal. Competing interests can arise in relationship to an organization or another person. Please follow this link to our website for more details on competing interests: http://journals.plos.org/plosone/s/competing-interests

Response: 

Our updated Competing Interests statement is as follows: “Cancer Epidemiology Research Programme runs some research projects funded by MSD and Seegene. Raquel Ibáñez has collaborated in a consultancy for Hologic and received HPV tests free of charge by Roche. Francesc Xavier Bosch has received travel/speaking grants from MSD, Seegene and Roche. The rest of the authors reported having no conflict of interest. This does not alter our adherence to PLOS ONE policies on sharing data and materials”.

7. Please amend the manuscript submission data (via Edit Submission) to include author Belén Lloveras.

Response: Thank you very much. It must have been a mistake. We have added the co-author Belen Lloveras. 

8. Please amend your list of authors on the manuscript to ensure that each author is linked to an affiliation. Authors’ affiliations should reflect the institution where the work was done (if authors moved subsequently, you can also list the new affiliation stating “current affiliation:….” as necessary).

Response: Authors' affiliation has been completed and reviewed. 

 

Response to Reviewers: 

Reviewer #1: 

Major comments

1. There is a crucial error in Table 1: the cumulative incidence at 9 years is not reported for the HPV+/cyto+ group. It could be a typo, but the value is not reported neither in the text, therefore, I suggest a revision of all the statistical analysis.

Response: We apologize for this omission. In the HPV+ /cytology+ category, all women at risk developed CIN2+ or CIN3+ or had a hysterectomy during follow-up by year 5, so they were censored. No woman within this category was at risk for the period 5-9 years. The table has been updated, in the manuscript, for further understanding.

2. Please uniform the Kaplan-Meier figures with the time span reported in Table 1 (i.e. 1, 3, 5, 9 years in the horizontal axis). Add also censored women (i.e., drop-out without the event of interest) in the curves. It seems strange that the end of the line (last available data) for the 4 groups is not equal to 9 years. Maybe a complete 9-year follow-up has not been reached by some group of women? The time span used for the calculation seems to be months, please specify in the methods.

Response: We thank the reviewer for this observation. The figures have now been modified, adding the years on the X axis according to the periods of time in table 1. We have modified the Y axis to put it in % by 100 and not in % by 1 as it was originally, also, in concordance with Table 1. In the figures do not include the number of censored women as they are in Table 1. The follow-up time period was set up to a year periods up to 9 years. Finally, the time span used for the calculation is not in months, but in years. It was already specified in methods section and in the graph on the X axis . 

We have expanded the statistical analysis description on the methods section, based on the reviewer's comments as follows: 

“The percentage of cumulative incidence of histologically confirmed CIN2+ and CIN3+ lesions and its 95% confidence intervals (CI) as well as the cumulative adherence to perform a next test were estimated and graphically represented using the Kaplan-Meier curves at 1, 3, 5 and 9 years. Data are presented according to the combination of baseline results of cytology and HPV tests up to 9 years follow-up. Incidence rates were calculated in person-years per 100 women. Pairwise comparisons between cumulative incidence curves were evaluated using log-rank test. Also, annual incidence rates of CIN2+ and CIN3+ lesions were calculated at 3, 5 and 9 years, as the number of cases per 100 women/year, according to all combinations of baseline test results, with their 95% CI (supplemental material). In all cases, time was expressed in years. All statistical tests were two tailed, and p-values below 0.05 were considered statistically significant”.

3. Please better specify the difference between censoring for follow-up and censoring as it is usually intended in the Kaplan-Meier estimates. Please describe in details in the methods which are the events of interest: I suppose that HSIL/CIN2+ is an event of interest, but also death from cervical cancer should be one, and hysterectomy for cervical diseases (if possible).

Response: Thank you for the comment. We have now expanded this important information. The main event of interest was the development of CIN2+ or CIN3+ along the follow-up period including death of cervical cancer. 

In the methods section, it was specified that a woman was censored for follow-up if a CIN2+ lesion was identified, or when there was a surgical treatment for a CIN1 lesion, or in case of hysterectomy for non-cervical causes. We have expanded the text as follows:

“The main event of interest was the development of CIN2+ or CIN3+ along the follow-up period. CIN2+ result included all histologically confirmed diagnoses of CIN2, CIN3, adenocarcinoma and invasive cervical carcinoma. In CIN3+ result, CIN2 is not included. A variable called “final diagnosis” was created including three categories: 1) women who had developed an CIN2+ or CIN3+ throughout the follow-up period, 2) women with a diagnosis other than CIN2+ or CIN3+, and 3) women with only the baseline tests that were considered lost to follow-up because of missing data. A woman was censored for follow-up if a CIN2+ lesion was identified, or when there was a surgical treatment for a CIN1 lesion, or in case of hysterectomy for non-cervical causes. In these cases, the woman contributed time until the diagnosis of CIN2+ or until time of surgical treatment or hysterectomy. Remaining women or in case of death for causes other than cervical cancer, the contributing time was accounted for until the last recorded test”.

4. I wonder if in Spain you are able to distinguish between deaths due to ‘uterus n.o.s.’ and deaths due to cervical cancer. I know this is a challenging issue in several countries. The same questions arise with regard to the possibility of identifying the reason of hysterectomy.

Response: Thanks for the observation that had been a relevant issue when looking at routine death certificates several decades age but we think the issue is now minimized. In all instances we extracted the available information from pathology laboratory or recorded case of death. Uterus n.o.s is now a minor cause of death within deaths from cancer of the uterus. We did not have any that were uterine cancer or uterus n.o.s. There were 3 gynaecological cancer-related deaths: one died from endometrial cancer, one from ovarian cancer, and another one from squamous cell cervix carcinoma.

Hysterectomy cases were also identified through the medical history. If the hysterectomy was for cervical malignancy, the woman was censored at the time of diagnosis of the cervical pathology and not the time of the hysterectomy. Further we identified several cases of hysterectomy for uterine prolapse. In these cases, when the report of the hysterectomy piece was available, if the cervix was normal, the women contributed up to the time of the hysterectomy, and the same when the hysterectomy was caused by other cancers. Hysterectomies for ovarian and endometrial cancer were recorded in our study.

Minor comments

5. Please better specify what do you mean for “predictive value”, i.e., both positive and negative predictive values? 

Response: We apologize for this unclarity. Yes, we refer to both, since in our study HPV test was able to distinguish women with a higher risk of developing cervical pathology (positive HPV) and, however, in negative women, HPV test confirmed prolonged protection during the study period. 

6. Methods, lines 122-124 death and end of follow-up should be included among the censoring times.

Response: Thanks for the comment. We included an explanation in Statistics analysis part of the manuscript. Please see the paragraph included in the response of the question 3. 

7. Please add the total for each group in Table 1. 

Response: Please see the modification of Table 1 in our response comment to the question 1.

8. The study could be underpowered (from a statistical point of view) to detect incidence of cervical lesions among the subgroup of women with HPV-/Cyto+ at baseline (only 16 women). Therefore, the results (incidence =0) in this group should be considered with caution. Please add this limitation to the discussion.

Response: We agree with the reviewer and we are aware of the limited number of events. A paragraph in the limitations of the study was added with this appreciation: 

“The yield of disease among the subgroup of women with HPV-/cyto+, at baseline (only 16 women), should be considered with caution, due to the low number of women under this stratum”.

9. Pg. 10 lines 194-196. I do not understand the meaning of this paragraph.

Response: We apologize for it. We have modified the text as follows. We hope it is better understood:

“In 36.7% of women, there was no evidence of a new test during the 9-year follow-up period. Most of these women classified as "lost to follow-up" were women with both baseline negative tests instead of women with both positive results (38.7% vs. 4.2%, p-value <0.001)”.

10. pg. 11, lines 225-229. I do not understand the meaning this comparison between groups.

Response: We apologize if we were not clear enough. We aimed to emphasize that cytology did not add any risk differences in the incidence of CIN2+ provided women were HPV positive. The 5-year incidence of the double positives at baseline was very similar to those HPV positives and cytology negative, emphasizing that it is the HPV test that defines the risk and not the cytology. 

Finally, the paragraph has been removed, as suggested by another reviewer to reduce the text of the results section.

Typos/language

Pg. 10: line 203, “3-years stablished” should be “3-year established”; line 212 “anormal”, should be “abnormal”

Pg. 11: Line 215 “CIN2+” should be “HSIL/CIN2+”; line 220 “cero”, should be “zero”; line 225 “statistical significantly” should be “statistically significant”

Pg. 12: Line 240: “baseline” 2 times; Line 241 “an HPV positive tests” should be “a HPV positive test”

Pg. 14: line 273 “accumulative” should be “cumulative”

Pg. 15: line 303 “To avoid detection of positive cases, that are unlikely to develop disease,” should be “To avoid detection of HPV positive cases that are unlikely to develop cervical diseases,”

Pg. 16: line 323 “least”, should be “at least”.

Response: Thank you for detecting the typos and edits that we have duly corrected.

 

Reviewer #2: 

1. I suggest to focus the results and the discussion on the specific issues related to the high risk group and in particular on the compliance to follow up and screening appointments in the following nine years. Now there is a paragraph in the results about timing of rescreening, maybe a figure with survival curves for having a test or the frequency of first and second test during follow up, separately for negative and positive women could be more informative.

Response: We appreciate the comment and we have modified our text. In fact, we have done a whole new analysis of cumulative adherence to perform a next cytology, HPV test, or biopsy, throughout the follow-up time stratified by baseline results. Tables and survival curves are now included. Also, in methods, results and discussion sections, we have included the data from this new analysis. The paragraphs are as follows, according to each section:

“Methods section: Cumulative adherence to perform a next test. Among women with HPV-/cyto- at baseline, any test performed after 180 days following baseline tests including HPV test, cytology or cervical biopsy were all taken into account. This was done to avoid including additional tests not related to screening that could have been performed before (e.g.: follow-up of infections, such as Candia Albicans, Trichomona Vaginalis, etc.). Among women HPV+/cyto-, tests performed at least 45 days after baseline were considered to not interfere with other tests performed concomitantly with the baseline ones. Among women HPV-/cyto+ and HPV+/cyto+, all subsequent tests performed, after baseline ones were considered without restriction. 

A woman was censured at the time that she had the next cytology, HPV test or cervical biopsy (event) registered after baseline according to the established criteria. The contribution time of that woman was time between baseline tests and the performance of next test. Women did no additional tests were censored on the date of baseline. In case of hysterectomy not due to cervical cancer, or death not related to cervical cancer, women was censored at the date of that event and considered of non-related event. If no date of death was available, the last date of woman's test was used. If none was available, it was considered the date of baseline.

Results section: 

Table 2 and figure 3 show the cumulative adherence to perform a next cytology, HPV test or cervical biopsy, along time by results of baseline screening tests. We observed that among women with both negative tests at baseline, in which the recommended interval for re-screening was after 3 years, only 40.5% had a next test registered during the 4 years from the baseline and 53.5% returned to perform a test along the following 6-years after baseline. Only 60.2% of these women had a next test recorded during the 9-years follow-up period, evidencing a poor adherence of these women to screening protocol recommendations. According to screening protocol recommendations, in HPV+/cyto- women, a cytology and colposcopy should be performed between 6 and 12 months or an HPV test at 12 months as follow-up. We observed that the 67.3% of these women had a follow-up test during the following 2 years from baseline, increasing to 74.5% before 3 years. However, in 16.3% of women there was no additional test recorded during the 9-year follow-up. Most women with a positive cytology at baseline and regardless of the HPV result had a follow-up test in the following year. 

Discussion section: 

In our study, we found that one third of total women did not return to screening after baseline tests. Among double negative women, only 40.5% had a re-screening before the next 4 years and 53.5% before the 6 years of follow-up, demonstrating that despite having participated in the screening activities, they did not meet the established intervals and most of those who returned, they did so at longer than recommended intervals. Unfortunately, we do not know if these women have searched care elsewhere. Under opportunistic screening approaches, loss to follow-up is common. We have shown that inviting these women can quickly revert the situation [36]. The use of personal reminders increased significantly screening participation even if only an invitation letter with a scheduled visit was used [36,37]. Generally, in organized programs, higher adherence to the screening programs is attained [37–39]. As well, women may perceive themselves to be at low risk of cervical cancer, and thus delay their attendance at screening as occurred in a Norwegian study [40]. A study carried out in Sweden also pointed out that women have a comprehensive rationale for postponing cervical screening, yet they do not view themselves as non-attenders [41]. 

On the other hand, a certain proportion of women, 17.2% of HPV positive and negative cytology and in 25% HPV negative women with a positive cytology, had no record of a second screening visit. A poor information or full understanding of a positive test could affect follow-up visit as has been seen elsewhere [42,43]. Good and comprehensive information remains a special need, particularly when the HPV test is positive, but the cytology does not detect any abnormality. The fact of having a negative cytology, associated with a lack of knowledge about the superiority of HPV tests in predicting disease, can give a false sense of security as observed also by others [40,41]. Some studies have shown that attendance at repeated tests is poor, particularly after a normal cytological test, reaching losses of 40% within the established one-year follow-up intervals [44,45]. Adequate communication is necessary to improve adequate management of screen positive women. For women who find it difficult to attend appointments, there may be substantial advantage to being able to self-collect at home”.

2. Also in the discussion, most of the issues afforded are general and many other studies and secondary literature already has treated these topics. What is specific of this study is the underscreened population over 40. I suggest to discuss points linked to this: I see two issues, but other can be interesting that I do not see: 1) compliance to follow up and acceptability; 2) co-testing is not recommended in Europe, but some researchers argue that for women not screened before 40, where the prevalence of cancer may be high, including cytology in the first screening episode could be useful, given that the cytology vs HPV relative sensitivity for cancer seems to be similar, cytology could add something. For the second point your study cannot give an answer, but you can confirm the high prevalence of cancers in this population, all HPV+/cyto+...

Response: We appreciate the thoughtful reviewer comment on how best direct the discussion. We agree that both points are critical in our data. 

Regarding point 1), we have modified the results text and discussion. Please see the paragraphs included in the response of the question 1.

Regarding point 2), according to the data from our study, cytology does not risk assessment in the co-test scenario. In our data, HPV test alone would provide the best risk estimate. This conclusion was already introduced in the discussion of the manuscript: “The inclusion of HPV test in addition to cervical cytology in women at risk of cervical cancer identified all CIN3+ cases but one, at year 9 since baseline, while cytology missed 6 cases of CIN3+. These results are consistent with our evaluation at 3 years and with the literature, supporting HPV as a standalone primary screening test, and that HPV testing is more accurate than cytology to predict disease also among underscreened women.”

Minor comments

3. Line 193 the lost to follow up seem different from those reported in the abstract.

Response: The abstract compared women with a negative co-test (HPV- and cyto-) with women with at least one positive result, either cyto and/or HPV test. However, in the results section, women with both tests negative are compared with those who have both tests positive at baseline.

Noting that it can be confusing, we have modified the information in the abstract to compare double positives with double negatives, as follows: 

“Lost to follow-up was higher among women with both tests negative compared to those with both positive tests (38.7% vs 4.2%, p-value <0.001)”.

4. Line 187 and line 194 the same results are reported with different indicators, thus it gives an impression of inconsistency and it is difficult to follow. I suggest to report the two measures in the same paragraph.

Response: As a new analysis of cumulative adherence to perform a next test has been done, these data have been modified. Please see the paragraphs included in the response to the question 1 part results.

5. Line 199 the average age at diagnosis is reported, but given the length of follow up it is difficult to manage this information. May be better to say how many in women below and over a certain age and to report it also for the whole cohort.

Response: We included the mean age at CIN2+ cases occur, and not by age group, since there were only 23 cases within an age range of 40 to 67 years. Stratifying the information further would result in groups with very low numbers and would not provide much additional information. 

However, below you have the table with the age at diagnosis of CIN2+ and the time (in months) between baseline test and CIN2+ diagnosis. If you think it may be interesting, we could include it in supplemental material. 

6. Lines 225-28: there is something not clear to me in the sentence: “Not statistical significantly differences were detected among women HPV positive and cytology negative at baseline at 9 years and women HPV positive and cytology positive at 5 years (24.6% CI:12.0-46.5; 32.6% CI:16.9-56.9 respectively).” Is the comparison between baseline or 9yy in HPV+/cyto- and HPV+/cyto+ at 5yy?

Response: We apologize if we were not clear enough. We aimed to emphasize that cytology did not add any risk differences in the incidence of CIN2+ provided women were HPV positive. The 5-year incidence of the double positives at baseline was very similar to those HPV positives and cytology negative, emphasizing that it is the HPV test that defines the risk and not the cytology. 

Bu finally, the paragraph has been removed, as suggested.

7. In general, I suggest to shorten the text of the results: tables and figures are much clearer…

Response: We have tried to reduce unnecessary text all over the manuscript.

8. Discussion: see general comments. 

Response: please see the paragraphs included in the response of the question 1 part results.

 

Reviewer #3: 

1. The data support the conclusions drawn although the sample size is relatively small - the authors should avoid overstatement/extrapolation from their data - e.g. while I accept that the risk of developing CIN3 in HPV negative women "regardless of cytology result at 5 years was 0" - this is likely to be influenced by power-limitations

Response: We appreciate the comment and a paragraph in the limitations of the study was added with this appreciation, especially emphasizing the number of women in the HPV-/cyto+ category. The new paragraph is the following: “The yield of disease among the subgroup of women with HPV-/cyto+, at baseline (only 16 women), should be considered with caution, due to the low number of women under this stratum”.

2. An important element of this work is that the cohort are "underscreened " women - we get to know what is meant by this in the introduction but underscreened is not defined in abstract, and arguably should be

Response: We apologize if we were not clear enough. We have added emphazised definition in the abstract as suggested. We have moved the definition to the objective part as follows: “To evaluate the 9-year incidence of cervical intraepithelial neoplasia grade 2 or worse (CIN2+) and cumulative adherence to perform a next test in a cohort of women aged 40+ years with no cervical screening cytology within a window of 5 years (underscreened women), after baseline cervical cytology and HPV tests”

3. A review of the English is recommended; there are some minor lapses in vocabulary and tenses eg line 83 "arouse" should be "arise" and for line 273 "accumulative" should be "accumulated"

Response: Thanks for appreciation. We have revised the language.

4. The data will be of importance locally and serve as key audit and service evaluation, with the potential to influence wider policy. However, as the authors describe - there are several larger published studies which describe longitudinal performance of HPV either as a stand alone test or as a co-test with cytology for the detection of cervical disease. The authors should try to describe the novelty/importance of these findings for the international community more clearly. 

Response: Overall, the article has been completed and greatly improved. One of the reviewers proposed a new analysis that was performed and included about cumulative adherence to perform a next test stratified by the baseline results. Now a new table and survival curve are also included. We believe that this new data puts more emphasis on underscreened women's behaviour in participating in screening programs and how HPV based screening can be better for these women with a high risk of developing cervical disease. We think now the discussion has improved. We have added the following new paragraphs below: 

“In our study, 86% of CIN2+ diagnosed among women those with both positive tests were detected during the first year and in most cases, screening tests were part of the diagnosis of CIN2+. In addition, having both positive tests generates a much higher risk during the first year than having one of them negative (27.0% vs. 6.5%). But above all, according to the data from our study, the risk increased if the positive test was the HPV test. Subsequently, the risk of diagnosis of these lesions decreases in later years. In fact, in our study, the cumulative risk of having CIN2+ lesions had a steep rise between the first 12 months and the third year, followed by stabilization or slow increase between the third and fifth year of follow-up. RCTs have also shown a reduction in detection of high grade lesions in the second rounds compared with the first round, highlighting the impact of using the HPV test in screening when it is compared with cytological screening [28,29]. This benefit of the HPV test could be critical for underscreened women that have persistently poor adhesion to recommendations. In our study, we found that one third of total women did not return to screening after baseline tests. Among double negative women, only 40.5% had a re-screening before the next 4 years and 53.5% before the 6 years of follow-up, demonstrating that despite having participated in the screening activities, they did not meet the established intervals and most of those who returned, they did so at longer than recommended intervals. Unfortunately, we do not know if these women have searched care elsewhere. Under opportunistic screening approaches, loss to follow-up is common. We have shown that inviting these women can quickly revert the situation [36]. The use of personal reminders increased significantly screening participation even if only an invitation letter with a scheduled visit was used [36,37]. Generally, in organized programs, higher adherence to the screening programs is attained [37–39]. As well, women may perceive themselves to be at low risk of cervical cancer, and thus delay their attendance at screening as occurred in a Norwegian study [40]. A study carried out in Sweden also pointed out that women have a comprehensive rationale for postponing cervical screening, yet they do not view themselves as non-attenders [41]. 

On the other hand, a certain proportion of women, 17.2% of HPV positive and negative cytology and in 25% HPV negative women with a positive cytology, had no record of a second screening visit. A poor information or full understanding of a positive test could affect follow-up visit as has been seen elsewhere [42,43]. Good and comprehensive information remains a special need, particularly when the HPV test is positive, but the cytology does not detect any abnormality. The fact of having a negative cytology, associated with a lack of knowledge about the superiority of HPV tests in predicting disease, can give a false sense of security as observed also by others [40,41]. Some studies have shown that attendance at repeated tests is poor, particularly after a normal cytological test, reaching losses of 40% within the established one-year follow-up intervals [44,45]. Adequate communication is necessary to improve adequate management of screen positive women. For women who find it difficult to attend appointments, there may be substantial advantage to being able to self-collect at home.”

Thank you very much for all the comments suggested by the editor and reviewers. We hope our manuscript is now more complete and suitable for publication and we look forward to hearing from you.

With kind regards,

Raquel Ibáñez Pérez

Information and Interventions in Infections and Cancer Unit. Cancer Epidemiology Research Program

Catalan Institute of Oncology. IDIBELL 

Gran Via de l’Hospitalet, 199 – 203. 08907 L’Hospitalet de Llobregat, Spain

Phone: +34 260 78 12 | Fax: +34 260 77 87 | E-mail: raquelip@iconcologia.net

---

## [Decision Letter · Decision Letter 1]

7 Aug 2020

Long-term protection of HPV test in women at risk of cervical cancer

PONE-D-20-10012R1

Dear Dr. Ibañez,

We’re pleased to inform you that your manuscript has been judged scientifically suitable for publication and will be formally accepted for publication once it meets all outstanding technical requirements.

Kind regards,

Maria Lina Tornesello

Academic Editor

PLOS ONE

Additional Editor Comments (optional):

Reviewers' comments:

Reviewer's Responses to Questions

**Comments to the Author**

1. If the authors have adequately addressed your comments raised in a previous round of review and you feel that this manuscript is now acceptable for publication, you may indicate that here to bypass the “Comments to the Author” section, enter your conflict of interest statement in the “Confidential to Editor” section, and submit your "Accept" recommendation.

Reviewer #1: All comments have been addressed

Reviewer #2: All comments have been addressed

2. Is the manuscript technically sound, and do the data support the conclusions?

Reviewer #1: Yes

Reviewer #2: Yes

3. Has the statistical analysis been performed appropriately and rigorously? 

Reviewer #1: Yes

Reviewer #2: Yes

4. Have the authors made all data underlying the findings in their manuscript fully available?

Reviewer #1: Yes

Reviewer #2: Yes

5. Is the manuscript presented in an intelligible fashion and written in standard English?

Reviewer #1: Yes

Reviewer #2: Yes

6. Review Comments to the Author

Reviewer #1: The authors have fully addressed all the Reviewers' comments. The paper has been improved and it is now suitable for publication.

Reviewer #2: Please check some typos in the new text, for example in line 273 I think "an" should be "and".

Line 153 and 355 "an CIN" should be "a CIN".

regarding answer to question 5, I think it would be useful to add these figures in the supplementary materials, this info could be used in meta-analyses.

7. PLOS authors have the option to publish the peer review history of their article (what does this mean?). If published, this will include your full peer review and any attached files.

Reviewer #1: No

Reviewer #2: **Yes: **paolo giorgi rossi

---

## [Editor Report · Acceptance letter]

17 Aug 2020

PONE-D-20-10012R1 

Long-term protection of HPV test in women at risk of cervical cancer 

Dear Dr. Ibáñez:

I'm pleased to inform you that your manuscript has been deemed suitable for publication in PLOS ONE. Congratulations! Your manuscript is now with our production department. 

Kind regards, 

on behalf of

Dr. Maria Lina Tornesello 

Academic Editor

PLOS ONE